# Crude Enzyme Concentrate of Filamentous Fungus Hydrolyzed Chitosan to Obtain Oligomers of Different Sizes

**DOI:** 10.3390/polym15092079

**Published:** 2023-04-27

**Authors:** Cleidiane Gonçalves e Gonçalves, Lúcia de Fátima Henriques Lourenço, Hellen Kempfer Philippsen, Alberdan Silva Santos, Lucely Nogueira dos Santos, Nelson Rosa Ferreira

**Affiliations:** 1Graduate Program in Food Science and Technology, Federal University of Pará, Belem 66075-110, PA, Brazil; 2Faculty of Food Engineering, Technology Institute, Federal University of Pará, Belem 66075-110, PA, Brazil; 3Faculty of Biology, Socioenvironmental and Water Resources Institute, Federal Rural University of the Amazon, Campus Belém, Belem 66077-830, PA, Brazil; 4Faculty of Chemistry, Institute of Exact and Natural Sciences, Federal University of Pará, Belem 66075-110, PA, Brazil

**Keywords:** chitosan, hydrolysis, filamentous fungus, enzyme

## Abstract

Chitosan is a non-cytotoxic polysaccharide that, upon hydrolysis, releases oligomers of different sizes that may have antioxidant, antimicrobial activity and the inhibition of cancer cell growth, among other applications. It is, therefore, a hydrolysis process with great biotechnological relevance. Thus, this study aims to use a crude enzyme concentrate (*C_EC_*) produced by a filamentous fungus to obtain oligomers with different molecular weights. The microorganism was cultivated in a liquid medium (modified Czapeck—with carboxymethylcellulose as enzyme inducer). The enzymes present in the *C_EC_* were identified by LC-MS/MS, with an emphasis on cellobiohydrolase (E.C 3.2.1.91). The fungus of the *Aspergillus* genus was identified by amplifying the ITS1-5.8S-ITS2 rDNA region and metaproteomic analysis, where the excreted enzymes were identified with sequence coverage greater than 84% to *A. nidulans*. Chitosan hydrolysis assays compared the *C_EC_* with the commercial enzyme (Celluclast 1.5 L^®^). The ability to reduce the initial molecular mass of chitosan by 47.80, 75.24, and 93.26% after 2.0, 5.0, and 24 h of reaction, respectively, was observed. FTIR analyses revealed lower absorbance of chitosan oligomers’ spectral signals, and their crystallinity was reduced after 3 h of hydrolysis. Based on these results, we can conclude that the crude enzyme concentrate showed a significant technological potential for obtaining chitosan oligomers of different sizes.

## 1. Introduction

Chitosan is a linear polysaccharide obtained by the partial deacetylation of chitin under alkaline conditions, which contains copolymers of d-glucosamine (deacetylated units) and N-acetyl-d-glucosamine (acetylated units) interconnected by β-glycosidic bonds (1→4) [1], and has the unique properties of biocompatibility, biodegradability, bioactivity, and non-toxicity.

In addition to the positive aspects of chitosan, it is necessary to meet technological conditions and specific needs that are enhanced depending on the size of the oligomer obtained through hydrolysis. The size of the oligomer is directly proportional to its respective molecular weight [2]. Based on molecular weight, chitosan can be grouped into low molecular weight (<100 kDa), medium molecular weight (100–1000 kDa), and high molecular weight (>1000 kDa) [3].

Compared to chitosan, its oligomers have a lower molecular weight, better water solubility, and greater physiological activities which involve: antimicrobial and antioxidant activity [4,5,6]; hypocholesterolemic properties [7]; antimutagenicity [8]; decrease in acrylamide formation in glucose/fructose-asparagine solutions [9]; and inhibition of tumor cell growth [10].

The standard method of depolymerization of chitosan is through acid hydrolysis. However, this method has some disadvantages, including the difficulty in obtaining oligosaccharides with a low degree of polymerization and in controlling the extent of hydrolysis, which often results in hydrolysates with a high amount of monosaccharides [11]. Furthermore, the reaction conditions require high temperatures and high concentrations of reagents, which often result in chemically modified oligosaccharides forming [12].

The use of enzymes in chitosan hydrolysis has received attention because it presents less variation in obtaining oligosaccharides than chemical hydrolysis [13]. In addition, the enzymatic method has advantages over chemical reactions because the enzymes act under milder conditions, have high specificity, and do not modify the structure of the glucose ring [14]. However, this method has the disadvantages of higher cost, limited availability of some enzymes, and slow action in viscous solutions, requiring a low substrate concentration and a more significant amount of enzymes [13].

Chitosanases and chitinases are specific enzymes responsible for the hydrolysis of chitosan and chitin, respectively. Chitosanases (E.C 3.2.1.132) are glycosyl hydrolases that catalyze the endohydrolysis of β-1,4-glycosidic bonds of chitosan. Chitinases can be classified according to their mode of action into endochitinases (E.C 3.2.1.14) and exochitinases (E.C 3.2.1.52), which catalyze the internal and external hydrolysis of chitosan [15].

Non-specific or promiscuous enzymes are also capable of hydrolyzing chitosan. These enzymes are cellulases, hemicellulases, pectinases, papain, pepsin, proteases, bromelain, and lysozyme, which are less expensive and can cleave the β-1,4-glycosidic bond of chitosan [5,16,17,18,19,20].

Cellulases are industrially essential enzymes and have traditionally been obtained from submerged fermentation by controlling extrinsic and intrinsic factors such as temperature and pH [21]. In this context, filamentous fungi are preferred for producing these enzymes rather than yeast or bacteria [22].

The search for new enzyme-producing sources and new enzymatic depolymerization mechanisms are objects of considerable scientific and technological interest. Thus, this study shows the possibility of using an efficient enzymatic extract obtained with few downstream steps capable of hydrolyzing chitosan to obtain chitooligosaccharides with varied sizes that present physiological activities.

## 2. Materials and Methods

### 2.1. Chitosan

Commercial chitosan (Polymar, Fortaleza, Ceará, Brazil) (cód. PB2112) (obtained from crustacean exoskeleton) with a certified degree of deacetylation of 85% was used.

### 2.2. Commercial Enzyme

Celluclast 1.5 L^®^ (Novozymes, Araucária, Paraná, Brazil), produced by Trichoderma reesei, was used as a standard in the hydrolysis of chitosan.

### 2.3. Microorganism

The filamentous fungus (code MIBA0664) belongs to the mycotheque of the Laboratory of Systematic Research in Biotechnology and Molecular Biodiversity of the Federal University of Pará (Brazil). This microorganism was isolated from a xylophagous mollusk using the procedure described by Ferreira et al. [23]. The strain was previously selected among others in the laboratory due to its potential endoglucanase activity.

### 2.4. Identification of the Microorganism

The fungal biomass used in the identification was obtained from cultivation in liquid medium Czapek Dox (Sigma-Aldrich, St. Louis, MO, USA, EUA) with the addition of 10% yeast extract and subsequent incubation in shaken flasks at 120 rpm/30 °C for five days. DNA was extracted according to the manufacturer’s instructions for the genomic DNA extraction and purification kit (Axygen^®^, ref. AP-MN-MS-GDNA-50) (Union City, CA, USA, EUA). The primer pair ITS-1f (5′-TCC GTA GGT GAA CCT GCG G-3′ and primer ITS-4r (5′-TCC TCC GCT TAT TGA TAT GC-3′) [24,25]. PCR was performed in a thermocycler (model TX96, Amplitherm^®^, State of São Paulo, Brazil) programmed at 95 °C/5′, 35 cycles of 94 °C/1′, 55.5 °C/2′, 72 °C/2′ (denaturation, annealing, and extension stages, respectively), and final extension at 72 °C/10′ [25].

Sequence annotation was performed using Geneious^®^ (version 9.1.5) and compared to GenBank. The sequence > 98% was considered to identify the filamentous fungus species.

### 2.5. Crude Enzyme Concentrate (C_EC_)

Initially, two culture media were evaluated for better enzymatic activity, established as endoglucanase activity [26]. Two media were used, modified Czapeck (C_zm_): carboxymethylcellulose (CMC) 10.0 g/L, sodium nitrate 3.0 g/L, potassium phosphate 1.0 g/L, potassium chloride 0.5 g/L, magnesium sulfate 0.5 g/L, and ferrous sulfate 0.01 g/L; and CPY: carboxymethylcellulose (CMC) 10.0 g/L, peptone 1.0 g/L, and yeast extract 20.0 g/L.

Cultivation was carried out in 500 mL conical flasks containing 250 mL of C_zm_ or CPY media. Ten discs of fungal mycelium fragments 5 mm in diameter were added and incubated in each flask at 30 °C, under constant agitation at 120 rpm. Endoglucanase activity was evaluated after seven days, and the assays were performed in triplicate.

After selecting the culture medium, a scale-up was performed, where five 1000-mL conical flasks containing 500 mL of C_zm_ and 20 mycelia discs (5 mm) were incubated under the same conditions used previously. Aliquots were withdrawn at 24-h intervals to assess endoglucanase activity. The final culture was filtered through a quantitative paper disc and a Buchner funnel under a vacuum to remove the biomass. The liquid phase was concentrated by lyophilization until 90% of the initial volume was reduced.

### 2.6. Protein Content

The amount of total protein present in the C_EC_ and the commercial enzyme was determined according to Bradford methodology [27] with modifications. Thus, 200 μL of C_EC_ or commercial enzyme was added to 1800 μL of Bradford reagent solution (0.1 mg/mL) with a reaction time of 5 min. The reading was performed in a spectrophotometer (Thermo, GENESYS, Fisher Scientific, Arendalsvägen, Göteborg, Sweden) at 595 nm. The blank was prepared under the same conditions, replacing the CEC with deionized water.

### 2.7. Endoglucanase Activity Assay (CMCase Activity)

The enzymatic activity was determined by quantifying reducing sugars [28], with modifications. In a 1.5 mL microtube, 500 μL of the enzymatic extract and 1000 μL of carboxymethylcellulose (CMC) at 0.5% diluted in sodium citrate buffer were added (0.1 M; pH 4.8). The reaction was carried out at 50 °C for 10 min in a Thermomixer (Thermomixer ^®^ compact, Eppendorf) at 400 rpm. After the established time, 100 μL of the reaction mixture was added to 200 μL of water and 300 μL of DNS reagent and incubated at 100 °C for 5 min. After the mixture was cooled, 1000 μL of distilled water was added. The blank control sample was evaluated under the same conditions, replacing the CMC with distilled water. The difference with the blank control sample determined the released reducing sugars (*Ra*). The reading was performed in a spectrophotometer at 540 nm. Enzyme activity was calculated using Equations (1) and (2):(1)$$Enzymatic\;activity\left(\frac{U}{mL}\right)=\frac{Ra\left(\frac{mg}{mL}\right)\times V_{r}(mL)}{T_{h}(\min)\times 0.18\times V_{e}(mL)}$$
(2)$$Enzymatic\;activity\left(\frac{U}{mg}\right)=\frac{Ra\left(\frac{mg}{mL}\right)\times V_{r}(mL)}{T_{h}(\min)\times 0.18\times P_{c}(mg)}$$
where: *Ra*: released reducing sugars; V_r_: reaction volume; T_h_: hydrolysis time; V_e_: enzymatic extract volume; P_c_: Protein content.

### 2.8. Identification of Proteins in Crude Enzyme Concentrate

#### 2.8.1. Protein Precipitation

The enzymatic extract concentrated by lyophilization was subjected to precipitation [29]. Proteins were precipitated by adding methanol, chloroform, and water in a ratio of 1:4:3 (*v*/*v*/*v*), respectively. The solution was vortexed, followed by centrifugation at 12.000× *g* for 5 min and 25 °C. The upper phase was discarded, and then methanol was added in a ratio of 3 mL to 1 mL of the initial volume of the sample. The solution was centrifuged at 12.000× *g* for 5 min and 25 °C. The supernatant was discarded, and the pellet was dried at room temperature 25 °C. The obtained pellet was washed with 50 mM bicarbonate and ammonium in Amicon^®^ Ultra 3 K MWCO (MilliporeSigma in the U.S. and Canada) 5 times, with subsequent centrifugation at 12.000× *g* for 1 hour.

#### 2.8.2. Protein Digestion

After washing with 50 mM bicarbonate and ammonium in Amicon^®^Ultra 3 K MWCO (MilliporeSigma in the U.S. and Canada), protein digestion was performed [30]. Each protein sample was homogenized with 50 mL of ammonium bicarbonate (50 mM), 10 mM DTT (dithiothreitol), and 0.25% RapiGest SF (Waters Corp., Milford, MA, USA). Digestion occurred from incubating trypsin samples with protein (1:50) at 37 °C for 16 h. Possible interferences in the samples were removed with the addition of formic acid (0.1%) and incubated at 60 °C for 60 min. Insoluble detergents were removed from samples by centrifugation at 10.000× *g* for 30 min. The supernatant was collected and frozen at −80 °C for subsequent identification of proteins by LC/MS.

#### 2.8.3. Identification by LC-MS/MS

Proteins were identified using the nanoElute nanoflow chromatographic system (Bruker Daltonics, Bremen, Germany), coupled online to a mass spectrometer (Hybrid Trapped Ion Mobility Spectrometry (TIMS), model Quadrupole Time-of-Flight (timsTOF Pro 2, Bruker Daltonics, Bremen, Germany). An aliquot (1 µL) sample, equivalent to 200 ng of digested peptides, was injected into a Bruker FIFTEEN C18 column (1.9 µm, 150 mm × 75 µm), from Bruker. The column was coupled online with a CaptiveSpray ion source (Bruker Daltonik GmbH). A typical RP gradient (Solvent A: 0.1% AF, 99.9% H_2_O; Solvent B: 0.1% AF, 99.9% CH_3_CN) was established in a liquid chromatography nanoflow system and separated at a flow rate of 500 nL/min. The column temperature was maintained at 50 °C. The chromatographic run lasted 60 min (30% of Solvent B for 55 min and 95% at 56 min; maintained at this percentage of Solvent B for another 4 min). The temperature of the capillary ion transfer line was set to 180 °C. Ion accumulation and mobility separation were obtained with an input potential ramp from −160 V to −20 V within 123 s. During the acquisition, to enable the PASEF method, the accumulation parallel to the fragmentation of the ions, the precursor *m*/*z* and mobility information was first derived from a Tims-MS full scan experiment, with an *m*/*z* range of 100–1700. Monocharged precursors were excluded for their position in the *m*/*z*-ion plane of mobility, and precursors that reached the target value of 20,000 a.u. were dynamically excluded for 0.4 min. The operational mode of the TIMS-TOF, MS, and PASEF were controlled and synchronized with the aid of the instrumental control software OtofControl 5.1 by Bruker Daltonik.

#### 2.8.4. Data Processing and Search Parameters

Data processing, protein identification, and relative quantification were performed using PEAKS Studio Software, Version 10.6, Bioinformatics Solutions Inc., Waterloo, ON, Canada. The processing parameters included carbamidomethylation of cysteine with fixed amino acid modification, methionine oxidation, and N-terminal acetylation, which were considered variable variations. Trypsin was used as a proteolytic enzyme, with a maximum of 2 possible cleavage errors. The ion mass shift tolerance for peptides and fragments was adjusted to 20 ppm and 0.05 Da, respectively. A maximum false positive rate (FDR) of 1% was used for peptide and protein identification, considering at least one unique peptide for identification as a criterion. All proteins were identified with a confidence level ≥ 95%, using the PEAKS Software algorithm and searching within the *Aspergillus* sp. database, using the UniProt database (http://www.uniprot.proteomes/ (accessed on 10 December 2022)).

The identified proteins were grouped into functional categories according to their functions and molecular weights using the database (http://www.uniprot.proteomes/ (accessed on 11 December 2022)).

#### 2.8.5. Identification of Microorganisms Based on Peptides

The identified peptides were analyzed using a bioinformatics tool for metaproteomics (https://unipept.ugent.be/ (accessed on 11 December 2022)) to verify the taxonomy of the studied microorganism. After identifying the microorganism, the proteins identified were analyzed against the microorganism database (https://blast.ncbi.nlm.nih.gov/ (accessed on 11 December 2022)).

### 2.9. Enzymatic Hydrolysis of Chitosan

#### 2.9.1. Crude Enzyme Concentrate

Hydrolysis was performed according to the method by Roncal et al. [18], with modifications. Chitosan hydrolysis was performed using a 1% solution (*w*/*v*) in sodium acetate buffer (pH 4.5; 0.1 M). From the C_EC_, 0.14 mg of protein/g of chitosan was added. Thus, 1000 μL of *C_EC_* was added in 100 mL of sodium acetate buffer, then 1000 mg of chitosan was added and incubated at 45 °C under agitation (120 rpm) at different times: 1; 2; 3; 4; 5; and 24 h. The hydrolysis reaction was stopped by heating the mixture at 100 °C for 5 min to inactivate the enzyme. The content was adjusted to pH 7, resulting in a chitosan precipitate. The precipitate was filtered and dried at 60 °C.

#### 2.9.2. Commercial Enzyme

Hydrolysis was performed according to the method by Roncal et al. [18], with modifications. Chitosan hydrolysis was performed using a 1% solution (*w*/*v*) in sodium acetate buffer (pH 4.5; 0.1 M). From the commercial enzyme Celuclast 1.5 L, 0. 51 mg of protein/g of chitosan was added. Thus, 20 μL of Celuclast was added in 100 mL of sodium acetate buffer, then 1000 mg of chitosan was added and incubated at 45 °C under agitation (120 rpm) at different times: 0.5; 1; 2; and 3 h. The hydrolysis reaction was stopped by heating the mixture at 100 °C for 5 min to inactivate the enzyme. The content was adjusted to pH 7, resulting in a chitosan precipitate. The precipitate was filtered and dried at 60 °C.

### 2.10. Characterization of Chitosan and Oligomers

#### 2.10.1. Molecular Mass Determination

The molecular mass of chitosan (Qt) and hydrolyzed chitosan (Qh) was determined by viscometry according to the method described by Garcia et al. [31], with modifications. Qt and Qh samples were prepared with a concentration of 0.005 g/mL in acetic acid buffer solution (0.3 M) and sodium acetate (0.2 M), pH 4.5, and kept at 30 °C, 120 rpm, for 24 h.

To determine the intrinsic viscosity, [η], the Qt and Qh solutions were diluted at concentrations of 0.004, 0.003, 0.002, and 0.001 g/mL, and the solution flow times were determined in a Canon Fensk capillary viscometer (Schott AVS 350) at 25 °C. The specific viscosity (η_sp_) was determined using Equation (3):η_sp_ = (*t* − *t*_0_*)*/*t*_0_(3)
where: *t* is the flow time of the chitosan solution and *t*_0_ is the flow time of the solvent.

The reduced viscosity (η_red_) was obtained through the relationship between specific viscosity and chitosan concentration (C), as in Equation (4):η_red_ = η_sp_/C(4)

Intrinsic viscosity [η] is defined as reduced viscosity, extrapolated to a chitosan concentration (C) of zero, as in Equation (5):[η] = (η_sp_/C)_c→0_ = (η_red_)_c→0_(5)

From the intrinsic viscosity, the molecular mass of chitosan was calculated using the Mark–Houwink equation, as in Equation (6):[η] = KM_w_^α^(6)
where: M_w_ is the viscosity average molecular weight, and K and α are constants that depend on the chitosan polydispersion and the solvent system used. The values of these constants were previously determined to be K = 0.074 and α = 0.76.

#### 2.10.2. FTIR Deacetylation Degree

The Qt and Qh samples were analyzed in an infrared spectrometer (Agilent, modelCary 360, Santa Clara, CA, USA, EUA) with Total Attenuated Reflectance (FTIR-ATR) and zinc selenide crystal (ZnSe), in the range of 4000 cm^−1^ to 650 cm^−1^, resolution of 4 cm^−1^ and 32 scans.

The degree of Qt and Qh deacetylation was determined by FTIR-AT by calculating the areas of the infrared spectral bands corresponding to the functional groups of amine (1350 cm^−1^) and CH_2_ (1465 cm^−1^), according to the methodology described by Barragán et al. [32]. The areas were calculated by the Spectragryph software (v. 1.2.14/2020) using the integration with baseline function [33]. The degree of acetylation (DA) was determined using Equation (7):A_1350_/A_1465_ = 0.3822 + 0.0313 GA(7)
where: DA: degree of acetylation; A_1350_: area under the curve of the infrared spectrum band with a wavenumber of 1350 cm^−1^; and A_1465_: area under the curve of the infrared spectral band with wavenumber 1465 cm^−1^. Values 0.3822 and 0.0313 were obtained by linear regression [34].

The degree of deacetylation (DD) was determined using Equation (8):DD = 100 − DA(8)

## 3. Results and Discussion

### 3.1. Identification of the Microorganism

The strain was identified as *Aspergillus* sp. (GenBank ID accession number MT135987) with greater than 97% similarity. In several studies, it was observed that the genus *Aspergillus* stands out in the production of enzymes that act in the hydrolysis of chitosan, such as cellulases [35], hemicellulase [18], and pectinase [36] by *Aspergillus niger*; chitin deacetylase by *Aspergillus nidulans* [37] chitosanase by *Aspergillus fumigatus* [38].

After identifying the enzymes in the secreted extract of *Aspergillus* sp, it was possible to perform a metaproteomic analysis based on peptides. The analysis allowed the identification of the microorganism belonging to the species *A. nidulans* (Figure 1).

### 3.2. Selection of the Culture Medium

The structural similarity between cellulose and chitosan—d-glucose polymers linked by β-1,4-glycosidic bonds—enabled carboxymethylcellulose to be an inducing substrate in both culture media (C_zm_ and CPY) in order to produce enzymes, which will later be used in the hydrolysis of chitosan. Figure 1 shows that the microorganism produced enzymes capable of hydrolyzing carboxymethylcellulose (CMCase activity), indicated by endoglucanase activity.

Among the culture media studied, the C_zm_ showed the highest enzymatic activity (Figure 2). In addition, an increase in enzymatic activity is observed up to the eighth day of cultivation, followed by a decrease in subsequent days, defining the eighth day in C_zm_ medium for the production of C_EC_.

### 3.3. Crude Enzyme Concentrate (CEC)

Table 1 shows the total activity (U/mL) and specific activity (U/mg) for the crude enzyme and the crude enzyme concentrate, respectively. The crude enzyme produced by the microorganism in C_zm_ medium during eight days of cultivation showed a specific enzymatic activity of 13.3 U/mg. After concentration by lyophilization, the crude enzyme showed an activity of 15.9 U/mg. The total protein content increased eight times after lyophilization, and there was a 19% increase in the specific activity. There was no significant increase in the specific activity. This may be related to the fact that no purification step was performed, only concentration, which enables more significant competitive inhibition by the saccharides present in the medium. Otherwise, this culture was characterized by low protein production, which contributed to good results of a specific activity.

### 3.4. Identification of Proteins in Crude Enzyme Concentrate

The fungus was grown in the C_zm_ culture medium added with chitosan. The proteins secreted by the microorganism in the culture medium were precipitated and submitted to LC-MS/MS analysis to verify the possible secretion of enzymes involved in the hydrolysis of chitosan by the fungus. Specific enzymes involved in the hydrolysis of chitosan are called chitosanases. They carry out hydrolysis from the cleavage of glycosidic bonds s β-1,4 [39]. These enzymes are usually extracellular and microorganisms such as *Bacillus subitilis* [40,41], and *Janthinobacterium* [42] showed capacity in the production of chitosanases. Among fungi, the genus *Trichoderma* [43] and *Aspergillus* [44,45] have stood out in this production.

Enzymes for polymers structurally similar to chitosan (such as cellulose and pectin) can also be used for the hydrolysis of chitosans, as they have chitosanolytic activity [5]. Regardless of whether it is a specific enzyme or not, chitosan is basically hydrolyzed by the action of glycosidic hydrolases (GH) [39]. In chitin and chitosan, acetyl groups remaining from GlcNAc play the nucleophilic role of the hydrolysis reaction by GH; thus, non-specific GHs can cleave the glycosidic bonds of chitosan [39].

Analysis by LC-MS/MS in the present study allowed for the identification of enzymes that hydrolyze chitosan. The peptides were blasted against the *Aspergilus* database. The taxonomic analysis based on identified peptides indicated that *A. nidulans* is the study microorganism (Figure 2); therefore, the identified proteins that did not belong to *A. nidulans/Emericella nidulans* were blasted against the *A. nidulans* database, which allowed the identification of several proteins, listed in Table 2. All identified proteins showed sequence coverage greater than 90% of *A. nidulans* proteins (Table 2). The accession numbers Q8NK02, Q5AUX2, Q5BA61, and Q9HGI3, which refer to the identification of *Emericella nidulans*, were not compared against *A. nidulans* because they are the same microorganism, only differentiated by the sexual or teleomorphic form.

Among the identified proteins were identified chitinases (E.C 3.2.1.14) and (E.C 3.2.1.52), responsible for catalyzing the cleavage of chitin and chitosan [46,47]. Depending on the type of cleavage, chitinases can be classified as endo- or exo-chitinases. Endo-chitinases (E.C 3.2.1.14) catalyze the internal hydrolysis of chains at random points along the polysaccharide, and hydrolyze the ends of the reducing or non-reducing polymeric chain, producing low molecular weight N-acetylglucosamine multimers [48], while exo-chitinases (E.C 3.2.1.52) act by randomly cleaving internal chitin sites. Exochitinases, in turn, are divided into two subgroups: chitobiosidases, which cleave chitin from the non-reducing end, releasing diacetylchitobiose (N-acetylglucosamine dimer); and 1-4-β-glucosaminidase, which cleave chitin oligomers released by endochitinases and chitobiosidases producing GlcNAc monomers. The β-hexosaminidase enzyme identified in the present work secreted by *A. nidulans* performs the hydrolysis of terminal non-reducing N-acetyl-D-hexosamine residues in N-acetyl-β-D-hexosaminides and can be used for the enzymatic synthesis of complex type sugar chains containing GlcNAc and GalNAc as components [48].

Chitinases are widely studied for different applications, such as biocontrol [49,50], as an antitumor agent [51], and bioethanol production [52], among others. The secretion of chitinases by *Aspergillus* spp. Is already described. It was observed that *A. flavus* secretes thermostable chitinases in a medium with high salt concentration [53], *A. niger* secretes thermostable chitinases that act in the biocontrol of *Candida* and *Galleria mellonella* [54], and *A. nidulans* secretes chitinases, cellulases, hemicellulases, esterases, and lipases when cultivated in sorghum straw [55].

In the pool of enzymes secreted by the microorganism, also identified were the enzyme cellobiohydrolase (E.C3.2.1.91), also known as exoglucanase. Two cellobiohydrolases were identified in the present work: Probable 1,4-β-D-glucan cellobiohydrolase A (Q5B2Q4) and Probable 1,4-β-D-glucan cellobiohydrolase B (Q8NK02). These enzymes release cellobiose units from cellulose’s reducing and non-reducing ends [56]. Cellobiohirolases can also hydrolyze chitosan, producing low molecular weight oligomers. In addition, it was verified that the hydrolysis product of chitosan through cellobiohydrolase is similar to that of chitosanase [56], with the formation of chitooligosaccharides that may have relevant bioactive effects, such as maintaining the immunological indices of patients undergoing chemotherapy [57], as well as antimicrobial and antioxidant activity [5].

Another identified enzyme was α-L-arabinofuranosidase (E.C 3.2.1.55). This enzyme releases L-arabinose and is involved in the hydrolysis of oligosaccharides and hemicelluloses [58]. Pectin lyase (E.C. 4.2.2.10), also identified, catalyzes the cleavage of the α1-4 glycosidic bond in pectic acid and pectin [59]. In addition to pectin hydrolysis, several studies have found that pectinases can be used in chitosan hydrolysis [20,60,61].

### 3.5. Chitosan Hydrolysis

The commercial enzyme (C_E_), Cellulast 1.5 L^®^, was used as a parameter to verify the efficiency in the hydrolysis of chitosan using the C_EC_ obtained in this study. C_E_ can cleave the β-1,4-glycosidic bond of chitosan [62], that is, the same region where chitosanase acts.

Figure 3 shows a decrease in molecular weight with increasing hydrolysis time for the two enzyme concentrates. The molecular weight of chitosan hydrolyzed by C_EC_ and C_E_ after 1 h was 85.20 and 71.14 kDa; at 2 h, it was 56.8 and 57.86 kDa, respectively.

Although the enzyme concentrates showed similar responses during hydrolysis, the amount of protein in C_E_ (0.51 mg of protein/g of chitosan) was higher than that of C_EC_ (0.14 mg of protein/g of chitosan). However, after 2 h of hydrolysis, there was no significant difference (*p* < 0.05) between the samples.

Due to the lower concentration of enzymes present in the C_EC_ compared to the C_E_, we can preliminarily infer that the C_EC_ showed greater efficiency in the hydrolysis of chitosan, highlighting the presence of endo-chitinases (E.C 3.2.1.14), exo-chitinases (E.C 3.2.1.52), and cellobiohydrolase (E.C 3.2.1.91) identified in this study.

### 3.6. Effect of Hydrolysis Time Using C_EC_

Figure 4 presents the molecular weight of chitosan after different hydrolysis times using the C_EC_. Chitosan with an initial molecular weight of 108.94 kDa showed a reduction of 47.80% (56.87 kDa) after 2 h of hydrolysis, 75.24% (26.86 kDa) after 5 h, and in 24 h, it was possible to reach a reduction of 93.26% (7.23 kDa). Thus, a significant decrease in molecular weight was observed, and thus, preliminarily a good correlation with the hydrolysis time to obtain the oligomers of interest.

We believe that the efficiency of C_EC_ is due to the presence of enzymes that have hydrolytic action, especially for obtaining chitooligosaccharides [46,56,62,63].

### 3.7. FTIR Analysis

FTIR analyzed the chemical structure of chitosan and its oligomers through the absorption bands (Figure 5). A pattern between the spectra with different hydrolysis times was observed at the 3435 cm^−1^ peak (a) due to the O–H stretching vibration merged with the N–H stretching band. The 2870 cm^−1^ bands (b) were attributed to the elongation of the C–H groups [4].

The spectra also showed characteristic bands for amide groups, including the characteristic band of C=O (amide I) elongation at 1648 cm^−1^ in the chitosan spectrum, with a displacement of this peak in hydrolyzed chitosan to 1660 cm^−1^ (c). At 1589 cm^−1^ (d), it showed bending vibrations of N–H coupled to stretching vibrations of C–N (amide II), and at 1330 cm^−1^ (e), characteristic of amide III is observed [7]. Furthermore, absorption bands were observed at 1150 cm^−1^, corresponding to the asymmetric stretching of the C–O–C bond (f). At 1067 and 1024 cm^−1^, the bands correspond to the vibration involving the C–O stretching (g) [64] and at 893 cm^−1^ (h), there is the absorption *β*-1,4 glycosidic linkages [16].

From the results, it is observed that the characteristics of chitosan were identified in the spectra, confirming its chemical identity before and after hydrolysis. In all spectra, the wavenumber values did not suffer significant displacements. However, other differences were observed between the spectra. Lower spectral intensities are observed in hydrolyzed chitosan because the functional groups vibrate with greater freedom of movement. The signal strength at 3465 cm^−1^ is due to more O–H groups, while the signal strength at 3364 cm^−1^ is associated with a more significant number of N–H units [4].

The absorption band at 3435 cm^−1^ shifted to a lower number of waves at 5 h and 24 h of hydrolysis, indicating that the crystalline order of chitosan was altered [16]. Furthermore, the absorption intensity relative to the C–H stretching band at 1380 cm^−1^ decreased at times of 3 h, 5 h, and 24 h, which indicated that intermolecular and intramolecular hydrogen bonds were weakened and its crystallinity was reduced [64].

### 3.8. Deacetylation Degree of Chitosan (DG)

The DG was determined using the FTIR spectra through the relationship between the integration values of the peaks at 1350 cm^−1^ and 1465 cm^−1^, according to Equation (8). This interaction showed the highest linear correlation compared to others in the infrared spectrum for different crustacean biopolymers [34]; Table 3 shows the DG for different hydrolysis times.

Table 3 shows a shift of 5 cm^−1^ in the values at the peaks of the integration areas. Thus, the values of 1345 cm^−1^ and 1460 cm^−1^ were considered and not 1350 cm^−1^ and 1465 cm^−1^ as recommended in Equation (8). Variations of this magnitude are likely to occur due to the degree of purity of the commercial sample compared to a standard sample. However, this fact did not influence the vibrational characteristics of the functional groups.

The DG values show no variation in polymer deacetylation when comparing chitosan before and after hydrolysis. This result confirms the idea of the specificity of enzymatic hydrolysis, in this case, in glycosidic bonds against conventional chemical hydrolysis with the possibility of obtaining unwanted values of GD.

## 4. Conclusions

The fungus used to produce C_EC_ was identified as *Aspergillus* sp. (GenBank ID accession number MT135987), and through metaproteomic analysis, the microorganism was identified as *Aspergillus nidulans*.

The enzyme concentrate produced by the fungus showed enzymes identified as cellobiohydrolases, chitinases, alpha-L-arabinofuranosidase, and pectin lyase. These enzymes, belonging to the glycosyl hydrolase class, show activity in the hydrolysis of chitosan, reported in other studies.

The C_EC_ showed greater efficiency in chitosan hydrolysis than the commercial enzyme. This is because, although the commercial cellulase has certified efficiency in the hydrolysis of cellulose, the concentrate may have presented a synergistic effect that catalyzed the hydrolysis of the inner and outer chain of chitosan.

The oligomers obtained by hydrolysis using C_EC_ showed lower absorbance of spectral signals due to the functional groups vibrating with greater freedom of movement. In addition, their crystallinity was reduced, and their degree of deacetylation was maintained.

The C_EC_ showed efficiency in the hydrolysis of chitosan to obtain oligomers of interest with few steps and lower costs.

## Figures and Tables

**Figure 1 polymers-15-02079-f001:**
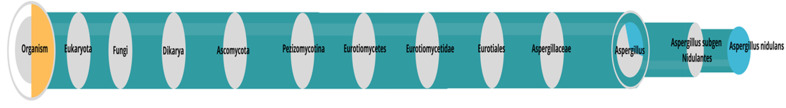
Dendogram based on peptides built by Unipept (https://unipept.ugent.be/, accessed on 11 December 2022). Phylogenetic tree based on the list of peptides identified using the lowest common ancestor (LCA) method.

**Figure 2 polymers-15-02079-f002:**
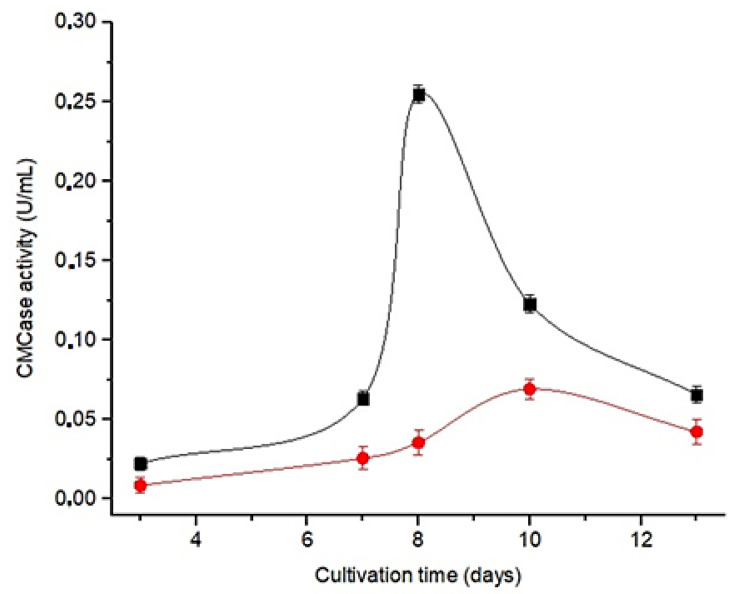
Enzyme activity profile (CMCase) in different culture media for *Aspergillus nidulans*: ■—modified czapek culture medium (C_zm_); ●—medium cellulose peptone yeast extract (CPY).

**Figure 3 polymers-15-02079-f003:**
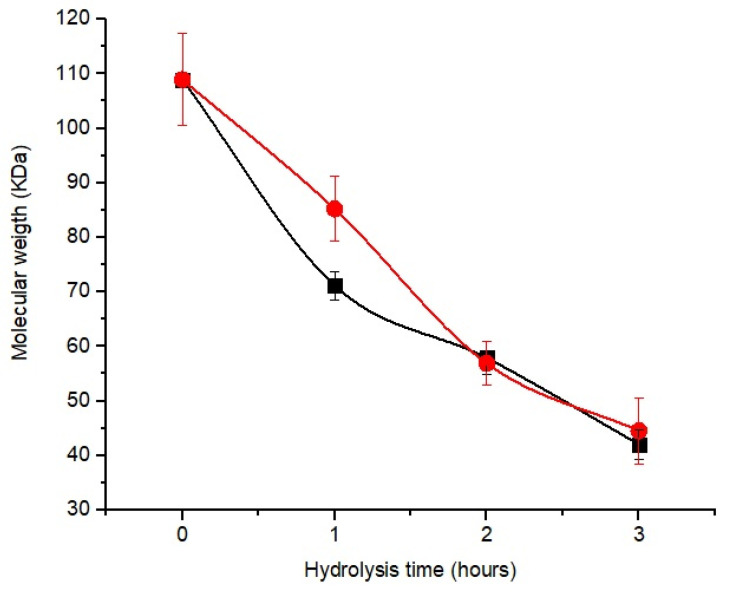
Chitosan hydrolysis profile to obtain oligomers as a function of reaction time and different enzyme concentrates: ■—C_E_ (commercial enzyme); ●—C_EC_ (crude enzyme concentrate).

**Figure 4 polymers-15-02079-f004:**
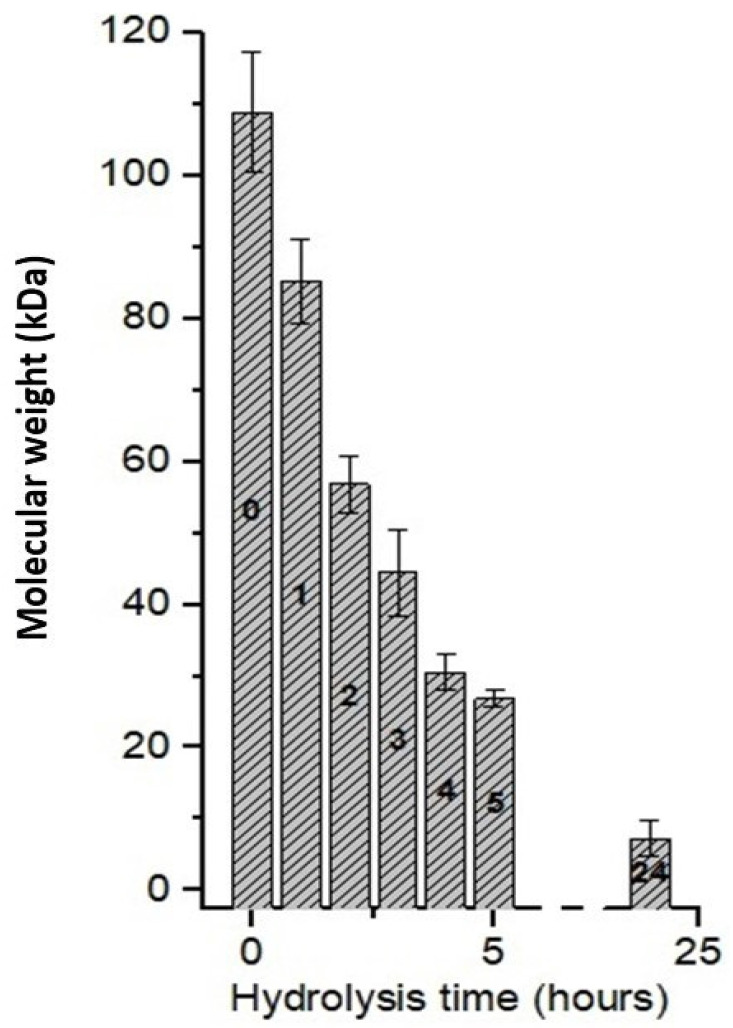
Chitosan hydrolysis as a function of reaction time using *A. nidulans* crude enzyme concentrate.

**Figure 5 polymers-15-02079-f005:**
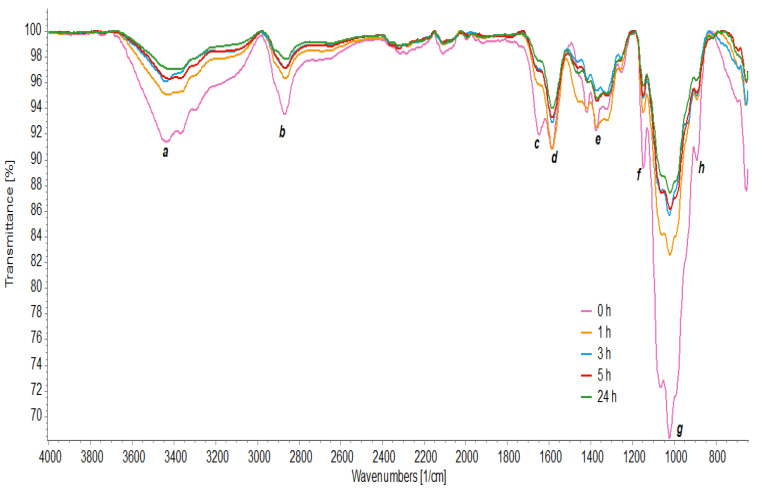
FT-IR spectrum of chitosan hydrolyzed with crude enzyme concentrate at times of 0 h, 1 h, 3 h, 5 h, and 24 h.

**Table 1 polymers-15-02079-t001:** Enzyme activity after extract concentration by lyophilization.

	Volume (mL)	Total Activity (U/mL)	Total Protein (mg/mL)	Specific Activity (U/mg)
Crude enzyme	500.0	0.22 ± 0.031	0.0165 ± 0.001	13.3 ± 0.97
Crude enzyme concentrate	25.0	2.23 ± 0.19	0.14 ± 0.013	15.9 ± 1.38

**Table 2 polymers-15-02079-t002:** Proteins identified by LC-MS/MS against the *Aspergillus* sp. and *A. nidulans* database.

Data on Proteins Identified by LC-MS/MS against the *Aspergillus* sp. Database					Data on Proteins Identified by LC-MS/MS against the *Aspergillus nidulans* Database		
Access Number—Proteins Identified against the *Aspergillus* sp. Database	Peptide	Mass	Ppm	*m*/*z*	Accession Number	Coverage (%)	Catalytic Class (E.C)
A1DMA5—Probable 1,4-beta-D-glucan cellobiohydrolase A OS = *Neosartorya fischeri*	VIANSVSNVADVSGNSISSDFC(+57.02)TAQK	26.692.603	16.7	8.907.712	Q5B2Q4—Probable 1,4-beta-D-glucan cellobiohydrolase A; flags: Precursor [*Aspergillus nidulans* FGSC A4]	99%	Glycosidase, Hydrolase (3.2.1.91)
Q8NK02—1 4-beta-D-glucan cellobiohydrolase B OS = *Emericella nidulans*	YGTGYC(+57.02)DSQC(+57.02)PR LNFVTQSQQK	14.625.605 11.916.248	0.9 −1.9	7.322.845 5.968.185	Q8NK02—1 4-beta-D-glucan cellobiohydrolase B OS = *Emericella nidulans*	100%	Glycosidase, Hydrolase (3.2.1.91)
Q5AUX2—Alpha-L-arabinofuranosidase axhA-2 OS = *Emericella nidulans*	ANSGATWTDDISHGDLVR	19.138.867	2.8	6.389.697	Q5AUX2—Alpha-L-arabinofuranosidase axhA-2 OS = *Emericella nidulans*	100%	Glicosidase, Hidrolase (3.2.1.55)
Q5BA61 Pectin lyase B OS = *Emericella nidulans*	SLVGEGSSGVIK	11.316.135	1.5	5.668.149	Q5BA61 Pectin lyase B OS = *Emericella nidulans*	100%	Liase (4.2.2.10)
E9QRF2 Endochitinase B1 OS = *Neosartorya fumigata*	IVLGMPLYGR	11.176.317	−0.6	5.598.228	G5EAZ3—Endochitinase B; AltName: Full = Chitinase B [*Aspergillus nidulans* FGSC A4]	90%	Glycosidase, Hydrolase (3.2.1.14)
Q9HGI3—Beta-hexosaminidase OS = *Emericella nidulans* OX = 162425 GN = nagA PE = 1 SV = 1	HISWGHSGPKPLSDVSLRTERDTDDSILTNAWNR	38.598.989	−5.7	12.876.329	Q9HGI3—Beta-hexosaminidase OS = *Emericella nidulans* OX = 162425 GN = nagA PE = 1 SV = 1	100%	Glycosidase, Hydrolase (3.2.1.52)

**Table 3 polymers-15-02079-t003:** Degree of deacetylation about hydrolysis time using crude enzyme concentrate.

Hydrolysis Time	Integrated Area (A_1345_)	Integrated Area (A_1460_)	DG (%)	CV (%)
Without hydrolysis	1.051	1.062	80	0.49
1 h	1.030	1.154	83	3.23
3 h	1.175	1.150	79	1.74
5 h	0.991	1.010	80	0.49
24 h	1.220	1.991	80	0.49

A_1345_: integrated area value for the 1345 cm^−1^ peak; A_1460_: integrated area value for the 1460 cm^−1^ peak; DG: degree of deacetylation; and CV: coefficient of sampling variation.

## Data Availability

The data presented in this study are available on request from the corresponding author.

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
