# Peer review of "Crude Enzyme Concentrate of Filamentous Fungus Hydrolyzed Chitosan to Obtain Oligomers of Different Sizes"

_polymers, 2023, doi:10.3390/polym15092079_

Round 1

Reviewer 1 Report

Reviewer’s Comments on Manuscript, polymers-2265158

            The study entitled “Crude Enzyme Concentrate of Filamentous Fungus Hydrolyzed Chitosan to Obtain Oligomers of Different Sizes is based on the hydrolysis of Chitosan (polysaccharide) into its oligomers using a crude enzyme obtained from the filamentous fungus.

The work presented in this manuscript is worthful. The introduction part is based on both old and new literatures related to this work. In the materials and method section, the methods used are standard ones. The result and discussion section is comprehensive. The references provided fully support the given text. However, the authors are required to incorporate the below mentioned minor changes in this manuscript before the manuscript is being onward processed.

1.     Natural environment is full of hundreds of microorganisms including various species of fungi. The moment they get a feasible environment they start growing. During the culturing of fungi, it is possible for the environmental fungi or the other microorganisms to have got contaminated in the main growing fungi. Thus, it will hard to believe the claim that the enzymes have been actually resulting from testing fungus (filamentous fungus). The authors are encouraged to answer what precautionary measures they adopted to exempt the possibility of the contamination of the unknown microorganisms from the environment.

2.     The authors are encouraged to put Ferreira et al. before [23] in Line-92 at Page-2 and Roncal et al. before [18] at Line-214 at Page-5 and similar changes are required throughout the manuscript.

3.     The names are not italic at various places, for instance, at Line-86, Page-2, the word Trichoderma reesei is not italic.

4.     The phylogenetic tree given in Figure-2 is completely invisible and difficult to read and understand.  The authors are encouraged to modify this figure and make it visible to be read out.  

5.     A proper reference is required in Line-125 at Page-3 after the word “Bradford's methodology”.  

6.     The authors are encouraged to put the original reference at the end of each protocol used.

7.     The authors are encouraged to briefly explain how the metaproteomic analysis enabled them to identify the microorganism. How they arrived at the conclusion to confirm that this is the only organism existing in the medium.

8.     The method is based on the application of the commercially available items. Which greatly effects the overall cost and utility of this method. The authors are encouraged to answer whether they have made any survey about the other possible cheap natural sources of the enzymes.

9.     Few references are out of format of this journal (Style of Page Number), for instance, Ref. 3, 4, 6,7,9 and so on. All the references need to be revisited for the possible corrections throughout.

I would like to recommend this manuscript for publication in this journal after incorporating the above small changes and answering the above queries.   

Author Response

Por favor, verifique o anexo.

Reviewer 2 Report

Thank you for the hard work. The reviewer have a couple of comments:

1. Have you compared this CEC with other commerical enzymes (other than Celluclast 1.5 L®?

2. Please add the statistical analysis and also n numbers to your figures

3. Have you done any economy analysis to compare the cost between your CEC and comerical enzymes (considering you are using higher dosage of 1000 ul of CEC 1000 ul vs 20ul of Celluclast )?

4. Any plans to investigate the optimal concentration of CEC? Also, please add a section of the future work

Reviewer 3 Report

 In general, the identification of proteins involved in polymer degradation is very poorly documented. We cannot verify the identification results obtained. I have rarely seen 100% sequence coverage in MSMS and proteomic techniques. In view of the theoretical tryptic peptide masses (above 7000 Da), the possible charge states of the ionised species and the MS experimental parameters of the type of chromatography used, the results must be provided. This is especially true when the sequence homologies for species close to  A. nidulans are greater than 90 to 95%. Before any acceptance, these results must be provided.

The SDS-PAGE electrophoretic profile of the precipitated proteins to assess the protein profile is of poor quality, even unusable, and the central part is largely unusable due to the presence of a possible halo of polymers or other contamination. This could be greatly improved. In addition, band sequencing should be undertaken directly by in-gel digestion to validate the authors' claims.

The authors present a study of the molecular weights of polymers after digestion by size exclusion chromatography (SEC), a technique commonly and generally used for the characterization of polymers.

Round 2

Reviewer 2 Report

NA

Reviewer 3 Report

The authors of the article" Crude Enzyme Concentrate of Filamentous Fungus Hydrolyzed 2 Chitosan to Obtain Oligomers of Different Sizes"have provided answers to the questions raised. The problematic parts have either been modified or removed as there were no results to support their hypothesis. I regret that some experiments, such as protein identification by in-gel digestion and MSMS, were not carried out as they are easy to perform. However, the manuscript in its revised form will be suitable for publication in Polymers.

Yours sincerely.